# Depth-Dependent Molecular Composition of Peat Organic Matter Revealed by ^13^C NMR Spectroscopy in the Mukhrino Carbon Supersite (Khanty-Mansi Region, Russia)

**DOI:** 10.3390/molecules30183663

**Published:** 2025-09-09

**Authors:** Evgeny Abakumov, Evgeny A. Zarov, Evgenii V. Shevchenko, Timur Nizamutdinov, Vyacheslav Polyakov, Elena D. Lapshina

**Affiliations:** 1Department of Applied Ecology, Saint Petersburg State University, Saint Petersburg 199178, Russia; e.shevchenko@spbu.ru (E.V.S.); timur_nizam@mail.ru (T.N.); slavon6985@gmail.com (V.P.); 2UNESCO CHAIR Department of Environmental Dynamic and Global Climate Changes, Yugra State University, Khanty-Mansiysk 628012, Russia; zarov.evgen@yandex.ru (E.A.Z.); e_lapshina@ugrasu.ru (E.D.L.)

**Keywords:** peatlands, soils, environment, ^13^C NMR, molecular structure, carbon monitoring

## Abstract

This study investigates the molecular composition of a Holocene oligotrophic sapric peat core from the Mukhrino Carbon Supersite in Western Siberia, with special reference to carbon pool stabilization mechanisms. Using ^13^C NMR spectroscopy and elemental analysis, we analyzed peat samples across depth gradients to assess organic matter transformation and stability. Results revealed distinct variations in aromaticity, hydrophobicity, and elemental composition with depth, indicating increasing organic matter stabilization in deeper layers. The study highlights the vulnerability of surface peat layers to mineralization under climate change, emphasizing the critical role of Siberian peatlands in global carbon balance. The findings provide insights into peatland carbon dynamics and underscore the need for conservation strategies to mitigate potential carbon emissions from these ecosystems

## 1. Introduction

The soils of Western Siberia (Russia) play a crucial role in stabilizing carbon pools, which is especially important under rapidly changing climate conditions. The approximate area occupied by peatlands is reported as 139,000,000 hectares [1], which constitutes about 8.14% of the total area of the Russian Federation. Up to 70–90% of peatland ecosystems are concentrated in Western Siberia. The West Siberian Lowland is the largest reservoir of carbon and peatland water. According to various estimates, the total carbon pool in Russian peatlands ranges from 100.9 to 118.9 billion Mg [1]. The gravimetric and volumetric contents of organic carbon in Siberian soils vary significantly across macrolandscapes [2] and within individual natural climatic subzones [3]. For instance, recent research demonstrates that the stocks of carbon were 50.2, 299.8, and 589.5 t/ha for podzol, histic cryosol, and oligotrophic histosols, respectively. Recently, it has been reported [4] that peat soils in Russia accumulate up to 29% of the organic carbon found in all Russian soils. Thus, the reserves of organic carbon in peat soils can exceed those in Podzols by 10–15 times and, when compared with thin leptosols, by tens or even hundreds of times.

There are many challenges in estimating organic matter reserves, most of which are related to field research methodologies and laboratory determinations [5]. However, this article focuses on analysis of the molecular aspects of organic matter stabilization in peat soils. Therefore, it is necessary to consider the molecular structural organization of organic matter. Molecular mechanisms of organic matter stabilization can be assessed using kinetic methods to study the emission and mineralization of organic matter in nature-like or laboratory experiments [3]. The use of spectroscopic methods for investigating both bulk peat material and humic acids (HAs) isolated from the same samples has been reported for peatlands in the northeastern part of the European region of Russia [6]. Detailed surveys of carbon species (groups of carbon atoms in organic molecules belonging to various chemical compounds) have been conducted for tundra soils in Alaska [7], the Yamal and Gydan peninsulas [8], permafrost-affected Chernozems in the southern part of Eastern Siberia [9], and even in Antarctica [10].

It has been shown that aromaticity negatively correlates with aliphaticity for various peat materials in Srednee Priobye (Khanty-Mansiysk region) [11], and aliphatic compounds prevail in the component composition of the peat. The author also suggested that the increased aliphaticity of HAs is caused by the woody origin of plant remnants. In contrast, grass peat materials may contain relatively higher proportions of aromatic carbon species [11]. On the other hand, the lignin of herbaceous plants may be more readily biodegradable due to its high cinnamic acid content, while the lignin of woody plants is characterized by a high guaiacyl/syringyl ratio, a property that contributes to its enhanced resistance to degradation.

However, the use of HA extracts to characterize the entire peat material is not always sufficiently informative, as different extractants can have varying effects on the molecular organization of HAs [12]. There are also studies on the use of nuclear magnetic resonance spectroscopy to identify peat in different zones of altitudinal mountain zonation. It has been established that peats at higher altitudes have a higher content of aliphatic fragments [13].

Solid-state ^13^C NMR spectroscopy has emerged as a powerful technique for characterizing intact peat organic matter, overcoming limitations associated with chemical extraction methods [14]. This approach provides quantitative data on carbon functional groups, while preserving the native molecular architecture—a critical advantage when studying complex, poorly decomposed peat matrices.

V.A. Kholodov et al. [15] reported ^13^C NMR spectra of HAs from an almost complete zonal sequence of soils on the Russian Plain, with the exception of polar soils. This work provided excellent confirmation at a molecular level of the theory of zonal humification by D.S. Orlov [16] and the zonal theory of soil-forming processes authored by V.V. Dokuchaev [17]. In this context, it should be emphasized that soils in the European part of Russia have been more intensively studied in terms of organic matter spectroscopy compared to Siberian soils. It should be noted that the extraction of humic substances with alkaline solutions can lead to alterations in their molecular organization. While HAs are highly representative of organic matter in mineral soils, they are not entirely representative of peats (histic materials). Therefore, proposals for recording spectra directly from entire soil samples [18] are also applicable to peats. The primary challenge lies in the high ash content of lowland peats, though this is less significant for oligotrophic peats. Another limitation of the investigation of organic matter in peat materials could arise due to their high iron or manganese contents, but in the case of oligotrophic histic materials this could have minimal effects.

While numerous studies have characterized bulk peat properties across Siberia [19], fundamental gaps remain in understanding the molecular-scale processes governing organic matter stabilization in these systems. In particular, the relationship between peat formation history (as recorded in stratigraphy) and its molecular signature remains poorly defined for ombrotrophic bogs of the West Siberian Lowland. The Holocene history of peat layer formation in Western Siberia has been investigated less than in the European part of Russia. That is why investigation of the molecular organization of the different age layers of histic soils may provide important information about the development of the organic matter system in a temporal sense.

Due to the limited knowledge about molecular mechanisms for stabilizing soil organic matter, particularly in the vast macro-area of Western Siberia, the objective of our work was to conduct ^13^C NMR spectroscopic studies of a peat bog core of varying ages at one of the most important monitoring sites in the region. To achieve this objective, we formulated the following aims: (1) to collect samples from a peat core, including dated samples from various depths, at a representative peatland plot, (2) to determine the C/N composition of the peat samples, and (3) to collect and interpret ^13^C NMR spectra of the peat materials.

## 2. Results and Discussion

### 2.1. Peat Macrofossil and Peat Properties Description

Previous studies have shown that the Mukhrino peatland was initiated by both terrestrialization and paludification approximately 10,000–11,000 calibrated years before present (cal. yr BP) [20]. The peat deposit at the study site exhibits clear stratification, reflecting shifts in plant communities and hydrological conditions throughout the Holocene. The bottom peat layer (400–500 cm) is formed by eutrophic plant remains, primarily *Equisetum*, sedges, and shrubs. The mesotrophic peat layer is thin, measuring only 10–20 cm, and consists of a mixture of *Sphagnum* mosses (*Sph. angustifolium*), grasses, and shrubs (*Eriophorum* and *Ericaceae* taxa). The upper layer of the peatland is composed of oligotrophic *Sphagnum* remains (dominated by *Sphagnum fuscum*) with an interlayer of oligo-mesotrophic peat, likely formed during droughts associated with the 4.2 kiloyear event [21,22]. The presented peat layer sequence reflects the natural succession of the peatland ecosystem from eutrophic to oligotrophic stages, consistent with data from other peatlands in Western Siberia [23]. Key characteristics of the peat material are presented in Figure 1.

In the studied profile, the upper layers, with a low 42% humification degree, exhibit high carbon lability due to their exposure to seasonal water level fluctuations and aeration. In contrast, the deeper horizons, maintained under permanent anaerobic conditions, preserve more stable forms of carbon despite their higher aromaticity. This underscores the importance of maintaining the natural hydrological regime of peatlands to minimize carbon emissions under climate change scenarios.

### 2.2. Elemental Composition

The percentages of key constitutional elements (C, H, N, and O) vary significantly with sampling depth, indicating differences in the origins of organic matter (Figure 2). The greatest variation in gravimetric concentration is observed for nitrogen. The first nitrogen peak occurs at depths of 70–80 cm and 130–140 cm, while a second, more pronounced peak is observed in the deepest layers of the column (420–430 cm and 470–490 cm). The lowest carbon content, approximately 42%, was found in the deepest layer of the soil section. Comparable low carbon contents have recently been reported for polygonal frozen bogs in Western Siberia [24].

The oligotrophic peat layers showed the lowest carbon (48.14 ± 1.86%) and nitrogen (0.64 ± 0.21%) concentrations, as well as the lowest bulk density (0.12 ± 0.03 g cm^−3^) and ash content (2.80 ± 2.27%). In comparison, mesotrophic peat exhibited the highest carbon content (53.27 ± 0.84%) and bulk density (0.24 ± 0.06 g cm^−3^), with intermediate nitrogen (1.09 ± 0.19%) and ash (2.80 ± 0.71%) concentrations. Eutrophic peat contained the highest nitrogen (2.29 ± 0.64%) and ash content (4.31 ± 3.74%), while showing intermediate values for carbon (52.15 ± 2.13%) and bulk density (0.22 ± 0.08 g cm^−3^).

The total carbon stock in the peat core reached 356.65 kg m^−2^, with oligotrophic peat contributing the largest proportion due to its dominance in the peatland profile. However, the highest carbon storage per 10 cm layer (14.02 kg m^−2^) occurred in mesotrophic peat, reflecting its greater bulk density and carbon content associated with abundant *Eriophorum* and *Ericaceae* remains. These values align with previous studies [22,24], though the total stock appears exceptionally high due to the core’s substantial depth. Total nitrogen storage amounted to 8.91 kg m^−2^, with maximum accumulation in mesotrophic (0.26 kg m^−2^ per 10 cm) and eutrophic peat (0.40 kg m^−2^ per 10 cm). The elevated nitrogen content in deeper layers likely reflects mineral admixture in these eutrophic deposits (Figure 2). In our peat cores, some horizons exhibit a maximum C/N ratio of up to 150, indicating that the organic matter is less enriched in nitrogen compared to the peat material investigated by Raudina and Loiko [24] in the permafrost-affected bogs of Western Siberia. C/N ratio, along with gravimetric carbon content, is considered an important predictor of CO_2_ emissions from peat soils affected by permafrost [25,26]. Thus, even within a single soil section (core), the quality of organic material can vary significantly, leading to different behaviors when exposed to surface conditions or changes in hydrological regime.

The H/C and O/C ratios (Figure 3 and Figure 4) are known indicators of the chemical composition and maturity of organic matter and are widely used in geochemistry and soil science [16]. The data in Figure 3 demonstrate that surface samples exhibit higher H/C values, consistent with a larger proportion of less transformed, aliphatic structural fragments. In contrast, the deepest layers are characterized by the lowest H/C and slightly elevated O/C ratios, which aligns with the ^13^C NMR data in showing an increase in aromatic carbon and a relative decrease in easily degradable alkyl chains. This evolution along the van Krevelen diagram is logical and expected for the humification and long-term diagenetic transformation of organic matter in peat deposits, reflecting processes such as dehydration, decarboxylation, and aromatization. The heterogeneity in the data points, both vertically and horizontally, is characteristic of peat soils in general [27]. This explains the significant variability in the chemical parameters of the soils studied. Additionally, fires, which regularly occur in both taiga and remote northern ecosystems, may have played a significant role in the content and stocks of organic carbon and nitrogen. In this context, layer-by-layer fluctuations in elemental composition may serve as indicators of changes in ecological and biogeochemical conditions [28].

Data on molecular ratios are provided in Figure 4. Although the data on elemental composition, particularly atomic ratios, are statistically significantly different for the peat layers, no correlation was found between these ratios and the type of vegetation composing the peat layers. Thus, the depth of the sample, the degree of organic matter transformation and the initial peat composition are the most important predictors of elementary composition in the peat core. At the same time, in the studied core, the C/N ratio is the most variable, ranging from 50 to 150, which is significantly higher than reported by Raudina and Loiko [24]. The observed decline in the C/N ratio in the deepest layers is a classic indicator of advanced organic matter decomposition, which is consistent with the presumed long-term anaerobic conditions (hydromorphism) in these horizons. Under such conditions, the preferential mineralization of nitrogen-poor carbon compounds (e.g., cellulose) over nitrogen-rich microbial biomass leads to a relative enrichment of nitrogen and a consequent decrease in C/N ratio. The increased enrichment by nitrogen in some horizons may result from the past wildfires and may have appeared during the development of the peatland ecosystem investigated. The observed decrease in oxygen content with depth, as reflected in the O/C ratios, can be explained by the preferential degradation of biomass constituents with the highest oxygen content, such as carbohydrates, compared to more recalcitrant, aromatic compounds like lignin or polyphenols. This selective preservation leads to a progressive decrease in the overall oxygen content of the organic material over time.

### 2.3. ^13^C NMR Spectroscopy of Peat Organic Matter

Information on the component chemical composition of organic matter is most effectively obtained using spectroscopic methods [22]. In this study, we interpret the ^13^C-NMR spectroscopy data shown in Figure 5. The interpretation of the spectra is provided in Table 1. The first observation from the spectra is that the degree of aromaticity of organic matter increases with depth. This is likely the result of dehydrogenation of organic matter, as indicated by the elemental composition data, and is also a consequence of humification. In this case, aromatization appears to be driven by proton loss rather than intensive oxidation of organic matter. At the same time, the decrease in oxygen may be expected from the selective degradation of carbohydrates, which represents the majority of plant biomass.

In general, a low degree of humification is characteristic of peats due to low aeration and high moisture content [29,30]. Peat strata are typically divided into layers with different levels of aeration, which significantly influence their chemical composition [31], especially in lower parts which are altered by the processes of diagenesis. The content of alkyl groups decreases significantly with depth, which is logical, as an increase in alkyl groups is accompanied by a decrease in the fraction of aromatic groups, and vice versa. Consequently, the AR/AL ratio expands from the surface to deeper layers, indicating an increase in the degree of organic matter stabilization in the lower horizons.

The results of ^13^C-NMR spectroscopy confirm that the peat deposits of Western Siberia contain substantial reserves of poorly decomposed organic matter, making them vulnerable to anthropogenic and climatic disturbances. From a peat science perspective, these findings can inform strategies for managing peatland ecosystems to preserve their carbon sequestration function. Additionally, the observed trends in molecular composition with depth may serve as indicators of past climatic events, such as droughts or periods of increased fire activity, thereby enhancing the potential for paleoecological reconstruction.

In soil science and humic chemistry, there is ongoing debate about the nature of soil organic matter and HAs in particular [32,33]. It is clear that HA isolates are not fully representative of soil organic matter as a whole. Therefore, our data, obtained from bulk samples rather than HA isolates, are particularly interesting and confirm that humification exists and can be quantified without isolating HAs.

Data on peat hydrophobicity (Table 1), calculated as AL h,r + AR h,r, show that hydrophobicity varies across layers. In addition to the nature of the organic matter itself, peat hydrophobicity is influenced by the composition of alkaline and alkaline earth cations [28]. Thus, total ash content and the cation composition of ash can affect the hydrophobicity–hydrophilicity of peat. E. Lodygin [34] reported that hydrophobicity for southern tundra soils is about 45%, increasing to 50% in southern taiga soils. In comparison, the hydrophobicity of our peat samples is lower. However, it should be noted that Lodygin [34] evaluated the elemental composition of humic and fulvic acids, whereas our work analyzes bulk peat samples.

The ratio of C,H-alkyl to O,N-alkyl, which characterizes the degree of decomposition of organic matter, is relatively low in the studied peats. This places them closer to fulvic acids, which are enriched in aliphatic carbon, and further from HAs, based on comparisons with previous results published by [29].

In summary, the levels of stabilization of organic matter in the peat core, as indicated by aromaticity, are relatively low but increase with depth. At the same time, the studied organic matter is characterized by relatively low hydrophobicity and a low degree of decomposition. A notable feature of the ^13^C-NMR spectra is their significant differentiation in terms of carbon species groups, reflecting the high diversity of humification precursors typical of bog ecosystems located at the junction of polar and boreal belts. It has been previously shown [30] that the degree of peat aromaticity depends on the rate of humification, with peats from bare patches exhibiting more humified organic matter.

Most of the current research on the carbon issue at monitoring sites in Russia is focused on studying the content and reserves of organic matter. However, stocks are not as important as the quality of organic matter at the molecular level. Thus, for the most famous carbon monitoring site in Western Siberia, the characterization of the organic matter of the peat core at the molecular level was carried out for the first time without extraction of organic molecules, i.e., in situ.

## 3. Materials and Methods

### 3.1. Regional Settings

The Mukhrino field station is located in the middle taiga subzone of Western Siberia (Russian Federation) [35] on the left second terrace of the Irtysh river (Figure 6). The study area is characterized by vegetation typical of the subzone, including coniferous forests (dominated by *Pinus sibirica*, *Picea obovata*, *Abies sibirica*, and *Populus tremula*), and ombrotrophic bogs (dominated by *Sphagnum* mosses and sedges) are presented for the studied territory.

General features of the peat core are provided in Table 2. Meteorological data were obtained from an automatic weather station located at the site. The monthly air temperature in June varied between 13.8 and 17.4 °C, and in January varied between −27.8 and 17.3 °C. The annual air temperature was −1.0 °C, and annual precipitation was 470 ± 68 mm, of which a quarter was snow.

The studied site is situated at the boundary between ridge and hollow ecosystems (60.891719 N, 68.675906 E) to capture variations in peat composition with depth formed by vegetation typical of these ecosystems. A peat core was extracted using a Russian corer (half-cylindrical shape, 50 cm length, 5 cm width, Eijkelkamp, Giesbeek, The Netherlands). The core was transferred to a C-shaped foamed polyethylene cassette, wrapped in plastic film, and stored in a freezer at −25 °C within 8 h. In the laboratory, the core was defrosted, and each 50 cm section was divided into five 10 cm samples. Each sample was further subdivided into three equal-sized subsamples. The first subsample was used for physicochemical analysis, the second for macrofossil analysis, and the third was stored as a reserve.

### 3.2. Basic Chemical Analyses

For physicochemical analysis, the subsample was oven-dried (48 h at 75 °C) and ground to a homogeneous state. Ash content was determined as the percentage loss of mass after combustion of approximately 1 g of dry peat at 525 °C for 8 h. Water holding capacity was measured gravimetrically after independent drainage of excess moisture in the laboratory. Carbon, nitrogen, and hydrogen content was measured using a CHN analyzer (EA3000 EuroVector, Pavia, PV, Italy). For this, 1–2 mg of a peat sample was combusted in a reactor filled with a catalyst (copper and chromium oxide) with an oxygen admixture in a helium flow. A chromatography column (EuroVector, Pavia, PV, Italy) was used for further separation, with detection conducted by a thermal conductivity detector (EuroVector, Pavia, PV, Italy).

The pH values were determined using a pH analyzer (Milwaukee Mi106 pH/Redox/Temp tester, Milwaukee Electronics, Milwaukee, WI, USA) via the potentiometric method. A van Krevelen diagram was constructed based on elemental analysis data, with oxygen content calculated using the difference between the three major elements. The results of elemental analyzes were corrected for ash content.

### 3.3. ^13^C-NMR Spectroscopy

For NMR, the peat samples were ground in a mechanical mill and sieved through a 1 mm mesh. They were then further ground in an agate mortar to achieve maximum homogenization. Deashing of peat samples was conducted in the laboratory by adding a hydrofluoric acid solution. Solid-state spectra of the peat material were obtained using CP/MAS ^13^C-NMR spectroscopy on a Bruker Avance 500 MHz NMR spectrometer (Bruker BioSpin GmbH, Rheinstetten, Germany) with a 3.2 mm ZrO_2_ rotor. The summarized data on the observed structural fragments within the peat materials are presented in Table 1. The chemical shifts were identified according to recommendations presented in paper [34]. Aromatic fragments were identified by the sum of chemical shifts at 105–144 ppm (unsubstituted aromatic carbon (H-Arom) and allyl-substituted aromatic carbon (C-Arom)), 145–183 ppm (O, N-substituted aromatic carbon (O, N-Arom)), and 184–190 ppm (quinone fragments (Arom=O)). Aliphatic fragments were identified by the sum of chemical shifts at 0–46 ppm (alkyl + α-amino (13 = methyl, 21 = acetate, 33 = polymethylene, 47–60 ppm (methoxy and ethoxy groups (O-CH_3_, O-CH_2_-CH_3_), O, N-substituted aliphatic fragments), 60–105 ppm (CH_2_OH groups of carbohydrate fragments, CHOH groups of polysaccharide rings, and esters (the signal of a carbon atom connected by a single bond to an oxygen atom in esters)), and 164–183 ppm (carboxyl groups, esters, and amides (COO-R)). Chemical shifts at 190–204 ppm were attributed to aldehyde and ketone fragments (C=O). The degree of hydrophobicity (AL h,r + AR h,r) was calculated as the sum of integrals at 0–47 ppm and 105–144 ppm. The degree of organic matter transformation (C, H-AL/O, N-AL) was determined by the ratio of integrals in the regions of 0–46 ppm and 46–110 ppm [34]. The spectra registration conditions were the following: contact time 2 ms, relaxation time 2 s, number of accumulations 3000–8000 scans.

### 3.4. Statistics

One-factor analysis of variance (ANOVA) was used for statistical data processing. Multiple comparisons were performed using the Tukey test with a single pooled variance (visualization of differences using compact letter displays). The confidence interval was set at 95% (α = 0.05). Data processing and visualization were performed using GraphPad Prism 10.3.1 (GraphPad Software, LLC, Boston, MA, USA).

## 4. Conclusions

This study analyzed vertical variations in organic matter structure across ten stratified peat layers from a key carbon supersite in Western Siberia’s peat deposits. Results demonstrate distinct elemental compositions correlated with layer age and origin, revealing a depth-dependent aromaticity gradient. Shallower carbon pools exhibit lower stability, indicating potential phased CO_2_ emission scenarios where surface layers mineralize before deeper, more recalcitrant fractions. Notable variations in hydrogenation/dehydrogenation degrees and oxidation reduction states of organic matter were observed across horizons. Bulk ^13^C-NMR spectroscopy confirmed low overall peat aromaticity, with a vertical increase in aromatic structures and reduced hydrophobicity compared to typical humic substances.

The bulk-sample NMR approach provides evidence that humification indices can be quantified directly in undifferentiated peat without extraction procedures, potentially offering a more representative assessment of native molecular architecture. Our findings suggest that the AR/AL ratio may serve as a useful indicator of decomposition states across trophic gradients in peat systems, though further comparative studies with other methods (for example pyrolysis gas chromatography/mass spectrometry of organic matter or Rock-Eval^®^ thermal analysis) would strengthen this interpretation.

Critically, the low hydrophobicity and aromaticity of studied peats render them potentially susceptible to mineralization under mechanical disturbance or climate warming, posing risks for organic matter reserves in northern Western Siberia. Paleoclimate signatures, including a 4.2 kyr drought signal in oligo-mesotrophic layers, provide molecular evidence of Siberian peatlands’ sensitivity to hydroclimatic shifts. The research underscores the necessity of accounting for vertical/horizontal heterogeneity in peatlands and historical biogeochemical impacts (e.g., fires and droughts) on organic matter stability. Layer-specific chemical variability highlights the need for high-resolution analyses when predicting carbon dynamics under environmental change.

## Figures and Tables

**Figure 1 molecules-30-03663-f001:**
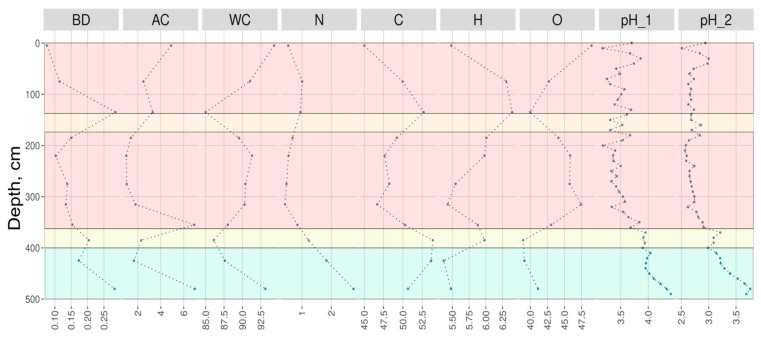
The main physical–chemical properties over the peatland depth. BD—bulk density, g/cm^3^; AC—ash content, %; WC—water content, %; N—nitrogen content, %; C—carbon content, %; H—hydrogen content, %; O—oxygen content, %; pH_1—pH of water extract; pH_2—pH of KCl extract. Orange—oligotrophic peat layer; yellow—mesotrophic peat layer; green—eutrophic peat layer.

**Figure 2 molecules-30-03663-f002:**
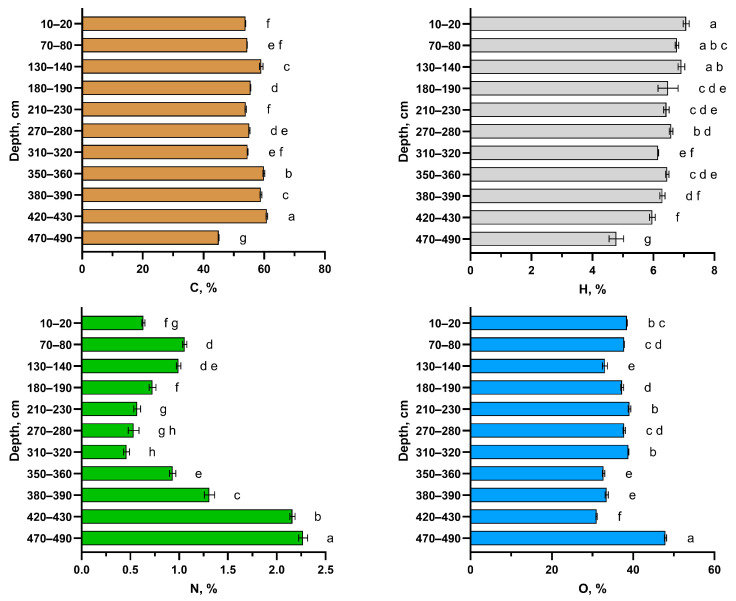
Elemental composition of HAs isolated from peat material calculated on ash content. a–h—compact letter display of ANOVA test, colors represents different elements content.

**Figure 3 molecules-30-03663-f003:**
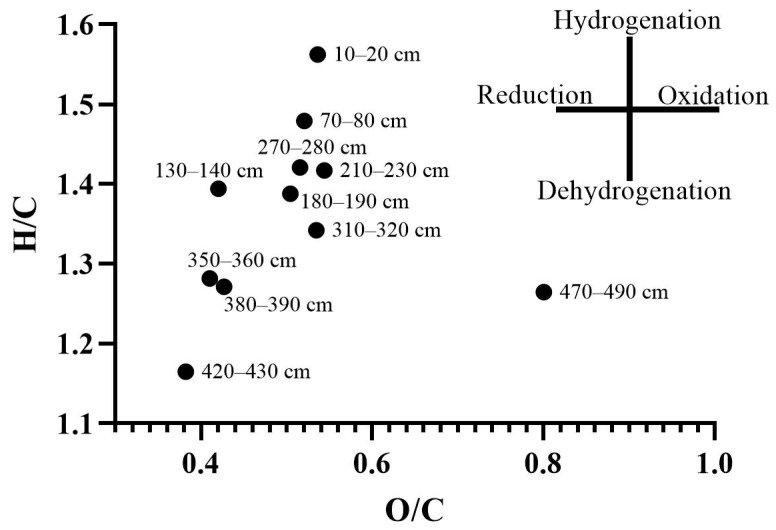
Van Krevelen diagram of elementary composition of HAs isolated from peat materials.

**Figure 4 molecules-30-03663-f004:**
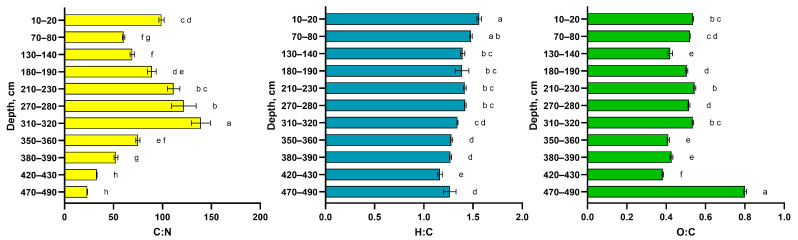
Atomic ratios in HAs isolated from peat material calculated on ash content. a–h—compact letter display of ANOVA test, colors represents different atomic ratios.

**Figure 5 molecules-30-03663-f005:**
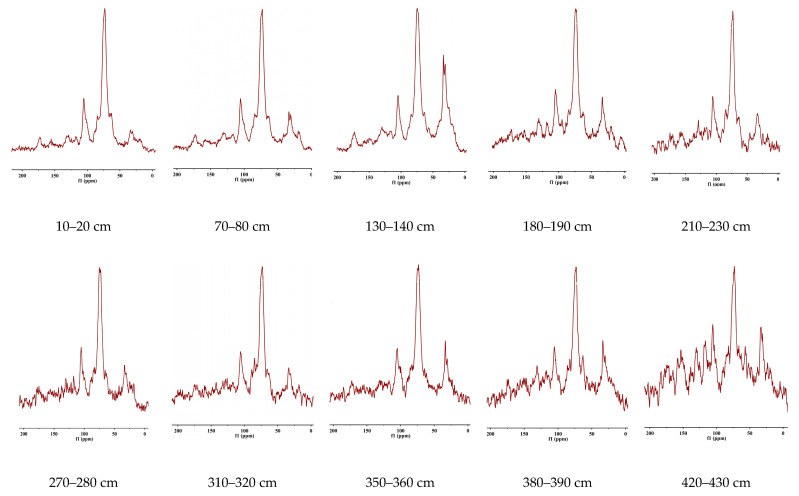
^13^C NMR spectra of HAs isolated from peat material with different sampling depths.

**Figure 6 molecules-30-03663-f006:**
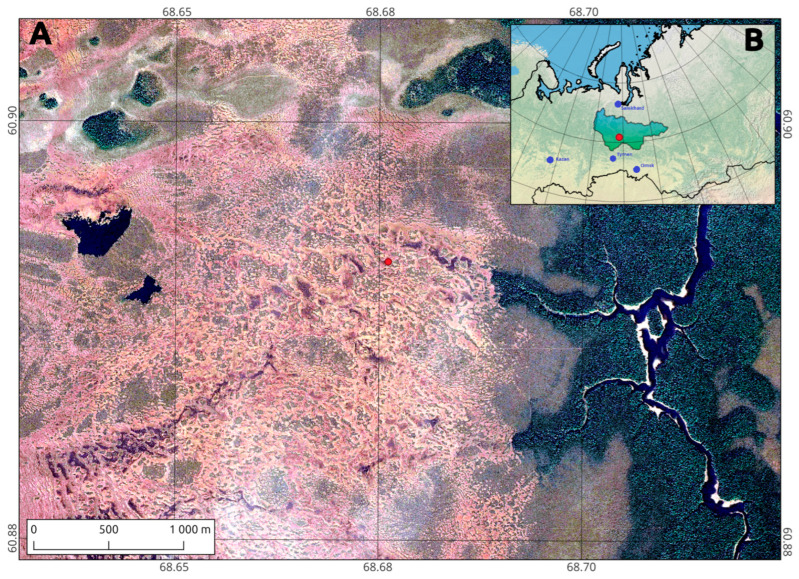
(**A**) Location of the core site; (**B**) Mukhrino field station (red point), Khanty-Mansi Autonomous Okrug-Yugra (area), and nearest major cities (blue points).

**Table 1 molecules-30-03663-t001:** Content of structural fragments of HAs isolated from the peat materials.

Sample Depth, cm	0–46	47–60	60–105	105–144	144–164	164–183	183–190	190–204	AL	AR	AR/AL	CH-AL/ON-AL	AL h,r + AR h,r
	Chemical shifts, % of ^13^C												
0–10	11.69	4.75	61.71	12.56	4.11	3.61	0.49	1.06	82.8	17.1	0.21	0.18	24.2
70–80	16.80	3.80	55.40	13.60	4.46	4.21	0.72	0.99	81.2	18.7	0.23	0.28	30.4
130–140	26.70	4.79	44.35	13.66	4.34	4.32	0.61	1.20	81.3	18.6	0.23	0.54	40.3
180–190	16.53	10.72	46.43	16.07	4.73	4.28	1.08	0.13	78.1	21.8	0.28	0.29	32.5
210–230	14.68	2.49	56.36	17.13	4.75	3.25	0.74	0.56	77.3	22.6	0.29	0.25	31.3
270–280	14.07	4.21	54.05	16.12	4.93	5.20	0.43	0.97	78.5	21.4	0.27	0.24	30.1
310–320	13.00	1.61	54.07	21.07	4.05	4.29	0.98	1.91	73.9	26.1	0.35	0.23	34.1
350–360	16.87	3.89	52.04	14.03	5.66	4.68	1.31	1.49	79.0	21.0	0.27	0.30	30.9
380–390	18.39	4.31	46.33	18.52	6.95	4.34	0.52	0.61	74.0	25.9	0.35	0.36	36.9
420–430	16.91	7.60	37.99	20.06	8.79	6.76	1.41	0.46	69.7	30.2	0.43	0.37	36.9
470–490	23.03	11.19	33.75	21.80	5.43	2.81	1.44	0.52	71.3	28.6	0.40	0.51	44.8

**Table 2 molecules-30-03663-t002:** The peat core features and compositions (average of three replications) of a single core column.

Peat Depth, cm	Plant Species	Ash Content, %	Water Holding Capacity, %
0–10	*Sphagnum fuscum*	4.91 ± 0.23	95.3 ± 0.5
70–80	*Sphagnum fuscum*	2.52 ± 0.08	88.6 ± 0.4
130–140	*Ericaceae* shrubs and wood remains	3.30 ± 0.12	84.2 ± 0.7
180–190	*Sphagnum fuscum*	1.43 ± 0.02	89.3 ± 0.6
210–230	*Sphagnum fuscum* + *Sphagnum angustifolium* admixture	1.00 ± 0.05	91.1 ± 0.4
270–280	*Sphagnum fuscum*	1.4 ± 0.07	90.4 ± 0.7
310–320	*Sphagnum fuscum*	1.80 ± 0.08	91.6 ± 0.9
350–360	*Sphagnum fuscum*	6.92 ± 0.07	88.0 ± 0.2
380–390	*Sphagnum angustifolium* with *Eriophorum* and *Ericaceae* shrubs	2.33 ± 0.04	85.2 ± 0.3
420–490	Horsetail and eutrophic mosses, sedges, and shrubs	4.00 ± 0.08	86.0 ± 0.3

## Data Availability

The data presented in this study are available upon the request from the corresponding author.

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
