# Peer review of "Depth-Dependent Molecular Composition of Peat Organic Matter Revealed by 13C NMR Spectroscopy in the Mukhrino Carbon Supersite (Khanty-Mansi Region, Russia)"

_molecules, 2025, doi:10.3390/molecules30183663_

Round 1
Reviewer 1 Report
Comments and Suggestions for Authors
Dear Authors,
Please find my recommendations listed below for "Depth-Dependent Molecular Composition of Peat Organic Matter Revealed by ¹³C NMR Spectroscopy in the Mukhrino Carbon Supersite (Khanty-Mansi region, Russia)" manuscript.
- L36: I recommend for authors to use standard taxonomic classification (World Reference Base or USDA Soil Taxonomy). "Oligotrophic Peat soils" is not a standard taxonomic classification. The correct terminology should be "Oligotrophic Histosols" or specify the exact WRB classification. Please consider this through the entire body of the manuscript.
- L39: "thin hyperskeletal soils" lacks also precision. lacks precision. The correct pedological term should be "Leptosols" (WRB) or "Entisols" (USDA), with specific reference to skeletal content thresholds
- L73-75: This statement should be sustained by examples/data and not finally reference (citation). Please avoid overgeneralization
- L77-78: Please avoid contradictory statements as they could induce confusions among readers. The manuscript sustain the necessity for bulk peat analysis while acknowledging that "humic acids are highly representative of organic matter in mineral soils, they are not entirely representative for peats."
- L80-81: Please consider also to refer at other critical limitations also. For example the paramagnetic interference from iron and manganese compounds, signals overlap in complex peat matrices, challenges in heterogeneous samples, etc.
- L82-85: Please be more specific. In my opinion the manuscript should identify precise mechanistic knowledge gaps rather than making broad statements about understanding limitations
- I recommend for authors that in introduction to consider also the temporal framework for peat formation processes. In my opinion this is critical to understand the molecular stabilization mechanisms in ombrotrophic systems
- L107: Please provide the units for ash. Include in table the standard errors or the standard deviation. Mention the number of sample replicates
- L108-110: Please mention the distance from ecosystem boundaries
- L110-113: Please justify why the authors choose to work with a single-core sample
- L119: Please check this "525°C for 8 hours". Up to my knowledge the standard protocol use 550 C and a longer time is considered
- Please include a subsection dedicated for statistical analysis
- L121: Please provide the combustion temperature
- L132-133: Please be specific and provide the concentration, contact time, etc...The authors should present also how they handled also the potential molecular structure alterations
- L146-148: Please provide the theoretical justification and include the uncertainties
- L133-145: Please include for the ¹³C NMR methodology the essential technical specifications (e.g., contact time for cross-polarization, line broadening parameters, number of scans, etc...
- L136-145: For chemical shift ranges please refer also at validation standards, peak deconvolution methods, etc.
- I strongly recommend for authors to reconsider the "Materials and methods" section, Please divide them into clear subsection. The methods description should contain all main parameters (procedural and instrumental) those that replication to be possible. Avoid a too narrative description style
- L151: The text implies Holocene-scale deposition for a 5-meter core without establishing accumulation rates or providing chronological constraints, making temporal interpretations scientifically invalid in my opinion
- L158: Please sustained the statement with radiocarbon dating evidence
- The authors should better sustain their results with relevant statistical metrics
- The manuscript should better consider the available literature in their discussions. In my opinion the manuscript provides descriptive observations without explaining underlying biogeochemical processes or molecular-scale mechanisms governing organic matter transformations. Also, the manuscript uses NMR results to validate the sampling approach while simultaneously using the sampling design to interpret NMR data, creating methodological circularity
- L295: The manuscript claims to have investigated "stocks of organic compounds" but didn't provides enough clearly the methodology for stock calculations. Please better consider this
- L301-303: Please reconsider this statement as in my opinion the data presented in the manuscript didn't sustain it enough well
- I recommend for authors to seriously reconsider the conclusion section also
Author Response
Dear reviewer!
Thank you for recommendations, provided, below you can see our replies and comments,
Sincerely yours,
Corresponding authors,
Evgeny Abakumov,
Professor, Dept. Applied Ecology, Saint-Petersburg State University, Russia
Dear Authors,
Please find my recommendations listed below for "Depth-Dependent Molecular Composition of Peat Organic Matter Revealed by ¹³C NMR Spectroscopy in the Mukhrino Carbon Supersite (Khanty-Mansi region, Russia)" manuscript.
L36: I recommend for authors to use standard taxonomic classification (World Reference Base or USDA Soil Taxonomy). "Oligotrophic Peat soils" is not a standard taxonomic classification. The correct terminology should be "Oligotrophic Histosols" or specify the exact WRB classification. Please consider this through the entire body of the manuscript.
Reply: thank you, the soil name corrected.
L39: "thin hyperskeletal soils" lacks also precision. lacks precision. The correct pedological term should be "Leptosols" (WRB) or "Entisols" (USDA), with specific reference to skeletal content thresholds
Reply: thank you, corrected
L73-75: This statement should be sustained by examples/data and not finally reference (citation). Please avoid overgeneralization
Reply: thank you, corrected.
L77-78: Please avoid contradictory statements as they could induce confusions among readers. The manuscript sustain the necessity for bulk peat analysis while acknowledging that "humic acids are highly representative of organic matter in mineral soils, they are not entirely representative for peats."
Reply: thank you, corrected.
L80-81: Please consider also to refer at other critical limitations also. For example the paramagnetic interference from iron and manganese compounds, signals overlap in complex peat matrices, challenges in heterogeneous samples, etc.
Reply: thank you, corrected.
L82-85: Please be more specific. In my opinion the manuscript should identify precise mechanistic knowledge gaps rather than making broad statements about understanding limitations
Reply: thank you, corrected.
I recommend for authors that in introduction to consider also the temporal framework for peat formation processes. In my opinion this is critical to understand the molecular stabilization mechanisms in ombrotrophic systems
Reply: thank you, corrected.
L107: Please provide the units for ash. Include in table the standard errors or the standard deviation. Mention the number of sample replicates
Reply: thank you, corrected.
L108-110: Please mention the distance from ecosystem boundaries
Reply: corrected.
L110-113: Please justify why the authors choose to work with a single-core sample
Reply: corrected
L119: Please check this "525°C for 8 hours". Up to my knowledge the standard protocol use 550 C and a longer time is considered
Reply: corrected
Please include a subsection dedicated for statistical analysis
Subsection was added (2.4)
L121: Please provide the combustion temperature
Reply: temperature 525 C is correct for comustions
L132-133: Please be specific and provide the concentration, contact time, etc...The authors should present also how they handled also the potential molecular structure alterations
Reply: information is modified.
L146-148: Please provide the theoretical justification and include the uncertainties
Reply: text has been modified.
L133-145: Please include for the ¹³C NMR methodology the essential technical specifications
(e.g., contact time for cross-polarization, line broadening parameters, number of scans, etc...
Reply: information has been added.
L136-145: For chemical shift ranges please refer also at validation standards, peak deconvolution methods, etc.
Reply: thank you, reference has been added.
I strongly recommend for authors to reconsider the "Materials and methods" section, Please divide them into clear subsection. The methods description should contain all main parameters (procedural and instrumental) those that replication to be possible. Avoid a too narrative description style
Reply: thank you, done.
L151: The text implies Holocene-scale deposition for a 5-meter core without establishing accumulation rates or providing chronological constraints, making temporal interpretations scientifically invalid in my opinion
Reply: thank you, done. Added the reference about the dates from the peat cores sampled in suburb of studied core.
L158: Please sustained the statement with radiocarbon dating evidence
Reply: The reference reflects the previous publication on the same peatland where shown that the layer of 4.2 kyr event was found as a properly humificaied and visible layer. See Tsyganov et al., 2021 and Railsback et al., 2018.
The authors should better sustain their results with relevant statistical metrics
Reply: Thank you, we have supplemented these aspects in the discussion.
The manuscript should better consider the available literature in their discussions. In my opinion the manuscript provides descriptive observations without explaining underlying biogeochemical processes or molecular-scale mechanisms governing organic matter transformations. Also, the manuscript uses NMR results to validate the sampling approach while simultaneously using the sampling design to interpret NMR data, creating methodological circularity
Reply: the text has been changed.
L295: The manuscript claims to have investigated "stocks of organic compounds" but didn't provides enough clearly the methodology for stock calculations. Please better consider this
Reply: the phrase is corrected.
L301-303: Please reconsider this statement as in my opinion the data presented in the manuscript didn't sustain it enough well
Reply: the phrase is corrected.
I recommend for authors to seriously reconsider the conclusion section also
Reply: corrected.
Reviewer 2 Report
Comments and Suggestions for Authors
A report for Molecules-3836039, entitled “Depth-Dependent Molecular Composition of Peat Organic Matter Revealed by 13C NMR Spectroscopy in the Mukhrino Carbon Supersite (Khanty-Mansi region, Russia)”
Authors: Typo in “Eelena Lapshina” (?)
(https://www.binran.ru/resources/current/herbaria/persones/1795-eng.html)
Abstract:
It is to be regretted that the summary did not provide more detailed information on the type of peat under study, including, for example, the degree of decomposition (sapric, hemic, fibric), or whether it was from a highland or lowland location, or whether it was oligotrophic or mesotrophic, or using classification from the USDA or FAO. In general, biogeochemists attach great importance to knowing whether peat has been formed from lignified vascular plants, or from vegetation lacking lignin, for example mosses.
Line 12: “ This study investigates the molecular composition and carbon stabilization mechanisms -- It is unclear whether this paper, which is primarily based on nuclear magnetic resonance (NMR), can provide any insight into "carbon stabilization mechanisms". Instead, the study investigates molecular composition e.g., "for its influence on carbon stabilization mechanisms", or similar.
Line 17: “ The study highlights the vulnerability of surface peat layers to mineralization under climate change,” -- In my opinion, this phrase constitutes an unnecessary endorsement of the alleged global change. The surface horizons of all peatlands are always more vulnerable to atmospheric agents because gas exchange with the atmosphere is more active there.
Introduction:
Line 35: Some scientific journals require the use of units of the international system (SI), for example, using Mg (megagrams) instead of tons.
Line 51: Please ensure that all references are cited in the text using consecutive numbers in brackets.
Line 64: “It has been emphasized that aromaticity negatively correlates with aliphaticity for various peat materials in Srednee Priobye (Khanty-Mansiysk region) and that 13C NMR spectroscopy is an informative tool for identifying peat genesis“ -- In my opinion, this sentence is both trivial and tautological. It should either be eliminated or replaced with something more appropriate. It is clear that the different forms of carbon can only be aromatic or aliphatic.
Line 56: “The author also suggested that increased aliphaticity of humic acids is caused by the woody origin of plant remnants. In contrast, grass peat materials contain relatively higher proportions of aromatic carbon species [10]. -- I have some reservations about the author's interpretation of the findings. I would therefore advise a certain degree of caution. It has long been hypothesised that the lignin of herbaceous plants is more readily biodegradable due to its high cinnamic acid content, which is linked to hemicellulose via ester bonds. Conversely, the lignin of woody plants has been found to exhibit a higher guaiacyl/syringyl ratio, a property that contributes to its enhanced resistance to degradation. It is my belief that the enhanced aromaticity detected in herbaceous vegetation may be attributable to factors other than the nature of the initial material. This is predicated on the premise that the biogeochemical cycle is more active due to the enhanced biodegradability of plant remains, which consequently leads to increased maturity after the accumulation of recalcitrant aromatic compounds. This is a standard occurrence when comparing forest soils with agricultural or deforested soils, where herbaceous remains predominate.
Lines 68 (and 302), (minor): Use M-dash instead of hyphen
Lines 70, 89, 93, 134… (minor) superscript in 13C
Lines 72 and 73 (minor): It may not be necessary to mention the initials in addition to the surname, especially when dealing with such well-known authors.
Line 67; “Due to the limited knowledge about molecular methods for stabilizing soil organic matter, particularly in the vast macroareal of Western Siberia,“ -- I would replace "methods" with mechanisms, processes, constraints…
Material and methods
Line 97: Repeated “conifer/ous…”
Figure 1: It would be advantageous to include the names of significant cities. This may also be of relevance to nations on the general map. This would be the case for major cities such as St. Petersburg, Moscow, Kazan and Omsk. Similarly, for map B, it is not explained what the blue area around the red circle and the elongated rectangle near Kazan are.
Table 1: Please indicate the units for all values given (e.g., %, or preferably (multiplying by 10) g/kg).
Please ensure that the N-dash is always used for numerical ranges.
Please use a decimal point instead of a comma.
Peat should be replaced with plant species.
Please note that Table 1 mentions the plant "angustifolium" on two occasions. It is evident that this refers to Sphagnum angustifolium, therefore it should be indicated by the genus and species names (with a lowercase letter) or abbreviated as S. angustifolium in subsequent instances. Similarly, on line 156, the genus name should be abbreviated as S., not Sph.
The same applies to Sphagnum fuscum, which is also referred to as "Fuscum" in the Table.
Please check the use of acronyms carefully: the term "humic acid" is used for the first time on line 48, where the acronym "HAs" should be introduced. However, throughout the manuscript, the complete words are used again on lines 57, 61, 70, 77, etc. Please note that the acronym is introduced on line 126, and the complete words are used again on lines 267, 270, etc.
Please refer to Table 1 for details on whether 'water content' refers to the maximum water-holding capacity at atmospheric pressure or to field humidity.
Lines 124 to 129: It is evident that oxygen is calculated using the difference between the three major elements.
However, the text does not state whether the percentages of these three elements have been corrected with respect to the ash-free sample. Although the ash content is not too high in these samples, if the percentages are not corrected with respect to the ash-free sample, the error accumulates in oxygen, and the percentages of the other three elements are also incorrect (underestimated).
Line 136: Please use the subscript in ZrO2. In the remainder of the page, where numerous functional groups or chemical radicals are indicated, the subscript numbers are also absent.
Line 339: “Aliphatic fragments were identified by the sum of chemical shifts at 0–47 ppm (methyl groups (CH3), methylene groups of long-chain alkyl chains (CH2), and methylene groups of branched alkyl chains (CH, C))” -- It is important to note that a common issue with 13C NMR is the overlap of signals produced by proteins with other significant regions of the spectrum. This overlap primarily occurs in the carbonyl and methoxyl groups, as well as the polymethylene alkyl groups. While the first two cases are referenced in this paper, the influence of peptides (α-amino) in the prominent alkyl group region is not indicated. I would describe this region, for example, as: "0–46 ppm = alkyl + α-amino (13 = methyl, 21 = acetate, 33 = polymethylene)".
Could you please indicate how BD (bulk density), AC (ash content %) and WC (water content %) were determined in the material and methods section, which focuses mainly on organic elemental analysis and 13C NMR?
Results and discussion
Line 157. Italicize Ericaceae (family name) as in Table 1.
Line 159: peat layers (?)
Line 162: characteristics
Figure 2 (caption): Although units such as g/cm³ or g/mL are also used to express density, kg/m³ is the SI base unit for density.
Line 169: “The hydrological regime is a key factor determining the degree of decomposition and preservation of organic matter in peat deposits. In the studied profile, the upper layers, despite their low humification degree, exhibit high carbon lability due to their exposure to seasonal water level fluctuations and aeration. In contrast, the deeper horizons, maintained under permanent anaerobic conditions, preserve more stable forms of carbon despite their higher aromaticity” -- I believe the information provided is accurate. However, the wording is very general. Peatlands are distinguished by their layered structure, with the surface material being the most recent and the deeper layers representing the oldest. All of the above is of great importance when it comes to justifying the properties, regardless of the water table.
Why “despite their low humification degree, exhibit high carbon lability“? -- Isn't it a contradiction? (despite).
Line 182: “The 182 lowest carbon content, approximately 42%, was found in the deepest layer of the soil section” -- Such low carbon contents have recently been reported for polygonal frozen bogs in Western Siberia [24]“ -- Indeed, the carbon percentage is slightly lower than expected, particularly in comparison to the upper horizons.
Despite the absence of any explicit mention to the contrary, it is assumed that the values for the four elements have been corrected based on an ash-free sample. Conversely, it would not be unexpected that the percentage of carbon (C) would be the lowest in the oldest horizon, where the ash content is the highest. Furthermore, the figure indicates that the oxygen content (or O+S+ash?) is higher in the deep horizon, where there is a greater quantity of ash.
Line 200 and below: “The elevated nitrogen content in deeper layers likely reflects mineral admixture in these eutrophic deposits. These authors also reported a C/N ratio variation of 39–85 (Figure 4). In our study, some horizons exhibit a maximum C/N ratio of up to 150, indicating that the organic matter in our peat column is less enriched in nitrogen compared to the peat material investigated by Raudina and Loiko [25] in permafrost-affected bogs of Western Siberia”-- In this paragraph, two important recommendations are put forward: firstly, the use of tautologies or trivial comments should be avoided; if the carbon-nitrogen ratio is higher, it is evidently because peat has less nitrogen.
Conversely, and with the requisite caution, hypotheses may be advanced regarding the potential correlation between variations in the C/N ratio and the distinct spatial or temporal impact of hydromorphism. It is important to recall that, in the French school's classification of classic humus types, “peat” and “anmoor” are studied separately from other humus types formed in terrestrial environments: The terms "mull", "moder", and "mor" are employed in this context. This phenomenon can be attributed to the consistently lower C/N ratios observed in hydromorphic soils compared to their terrestrial counterparts. In the former case, this does not indicate greater biological activity, but simply the fact that protein is less easily biodegradable in the hydromorphic environment.
Consequently, and always with caution, an attempt could be made to explain variations in C/N ratio by factors such as the different levels of the water table or the different seasonality of hydromorphism. It is important to note that other factors may also have had an impact, although it should be noted that these are entirely speculative and should not be mentioned without proof. One such factor could be the presence of fires in the past, which have been known to produce an increase in heterocyclic nitrogen forms. These forms are extremely resistant to degradation.
Lines 216 and below: “One might expect the degree of oxidation to increase with decreasing hydrogenation, but this is not the case. Instead, there is degradation of alkyl groups without the formation of oxidized groups, such as carboxyl groups“ -- On the one hand, the van Krevelen diagram would be discussed with reference to different proportions of alkyl, or aliphatic, and aromatic constituents, rather than hydrogenation or dehydrogenation processes. Conversely, it is not considered necessary to seek singular interpretations regarding the O/C ratios in the van Krevelen diagram. The evolution observed in the graph is logical and expected for humic-type materials with different degrees of maturity.
Indeed, the youngest material in the surface layers has a higher content of aliphatic constituents, and the material with greater degrees of maturity or diagenesis in the deeper layers is the one that presents greater aromaticity. The necessity to seek alternative explanations for the decline in oxygen content with depth is rendered moot by the classical trend towards maturity, discernible in van Krevelen diagrams, characterised by the progressive convergence of the coordinates of increasingly mature samples along the primary diagonal, converging towards the coordinate axis.
This trend is explained as the biomass constituents with the highest oxygen content are carbohydrates. As these carbohydrates degrade preferentially in comparison to lignin or polyphenols, which are aromatic in nature, the oxygen content of the resulting material progressively decreases.
On the other hand, it has been demonstrated that the process of humic acid formation is accompanied by an increase in carboxyl groups. However, in hydromorphic soils, this tendency is not necessarily evident in the van Krevelen diagram applied to whole peat material.
It has been extensively documented in classical literature, e.g., the classical book by Durand: Kerogen: An Insoluble Organic Matter Derived from Sedimentary Rocks. In addition to establishing a correlation between aliphaticity and aromaticity with maturity, he advances a series of other observations on the impact of hydromorphism, distinguishing between terrestrial and aquatic materials, as well as materials of mixed origin, once again based on their position in the van Krevelen diagram. (However, the application of this theory within the same deposit where the effects of hydromorphism and maturation time completely overlap remains unclear).
Finally, the O/C ratio in the sample at the greatest depth (470-490 cm) appears to be significantly elevated, as it also exhibits the highest ash content.
In the absence of verification confirming the presence of an error, which is not necessarily an experimental error but rather a calculation error, as previously hypothesised, this outcome would be anticipated. In the event of the carbon, hydrogen, nitrogen and oxygen percentages not being ash-free, it can be deduced that the used carbon, hydrogen and nitrogen values would be lower, whilst the percentage of oxygen would be exaggeratedly higher.
Figure 4: caption: Krevelen
Apparently, there are two Fig. 5.
Figure 5 (former): The term "turf" is used in the figure caption instead of "peat."
In addition, the decline of the C/N ratio is a very relevant indicator, as it is known to decrease progressively in the deepest layers, which are presumed to be most affected by hydromorphism. I do not consider it necessary to present graphs for the H/C and O/C atomic ratios in bar form. Furthermore, I always recommend checking that the values are calculated using an ash-free sample. If this information were available, many of the graphs presented could show the upper, lower, or average limits of the water table, if it varies substantially throughout the year (which, incidentally, is not very detailed in the text).
Line 232: “ This indicates that the depth 232 of the sample and the degree of organic matter transformation are more important than 233 the peat composition. ” -- It is my understanding that in this sentence, the authors are not referring so much to the composition of the peat as to the chemical composition of the original vegetation.
Line 245: “In this case, humification appears to be driven by proton loss rather than intensive oxidation of organic matter“ -- In line with my previous recommendations, I suggest using terms such as "aliphatic content" or "aromatic content," or "aromatization" or "selective degradation of aliphatic or O-alkyl constituents," where appropriate.
In addition, it is my professional opinion that the observed trends are indicative of classical peat maturation. The absence of a progressive increase in oxygen content should not be regarded as an abnormality. It is important to note that the process being studied involves working with complete peat and not with humic acids. The decrease in oxygen is expected from the selective degradation of carbohydrates, which represent the majority of plant biomass. This is particularly evident in peat dominated by Sphagnum, a lignin-lacking plant.
Figure 5 (latter): Improve the proportions of the lettering, especially in the x-axis labels.
Caption: 13C (with superscript) instead of 13-C
With regard to the spectra, I generally prefer to normalise them by height or area, although this is a relatively minor detail.
Table 2
Please use decimal dots instead of commas.
Please use N-dash for numerical ranges (see first column and first row for ppm ranges).
I consider it useful that, although it has already been briefly explained, some words be indicated to illustrate the meanings of the different chemical shift ranges, for example alkyl, O-alkyl, H-aromatic, C-N-aromatic...
Line 251 and below ” In general, a low degree of humification is characteristic of peats due to low aeration and high moisture content [30].” -- While the aforementioned points are accurate, it is advisable to exercise caution when writing, as there are two overlapping factors that influence maturity depending on depth (hydromorphy and time). The deep layers are the most affected by hydromorphy; however, they are also the oldest, and therefore exhibit the highest degree of diagenesis.
Line 291: “This suggests that if peat layers currently not exposed on the surface were to become exposed, active humification would likely occur“ -- Whilst there may be some veracity in this claim, it is a rather general comment that would benefit from further clarification. It may be preferable to avoid mentioning this issue, given that peats are hydromorphic soils where the degree of humification can reach advanced levels after several centuries without being exposed to active gas exchange with the atmosphere. The formation of ITS humic substances is a gradual process that requires a significant amount of time.
Following peatland desiccation and the lowering of the water table, the degree of humification typically increases. However, mineralisation predominates, with peats spontaneously combusting without flame. This process has destructive yet intense effects on the composition of the peaty material.
Conclusions
Line 300: “Given the demonstrated lability of surface peat layers and their disproportionate sensitivity to temperature fluctuations our molecular characterization suggests these systems may exhibit nonlinear responses to warming” -- Please be advised that the following applies: I would advise against making speculative statements regarding the alleged effect of temperature. The effect of temperature is highly dependent on humidity, particularly in relation to whether the temperature increases during the cold or warm seasons of the year. However, the primary determining factor is the duration of the desiccation periods of the surface layers. Furthermore, the potential changes in vegetation types over time, in combination with the historical impact of fire or the contribution of surface materials, significantly complicate the reconstruction of the processes of organic matter formation in peatlands. This suggests that conclusions should be drawn with caution.
Line 311: “(1) humification indices can be reliably obtained without alkaline extraction“ -- I consider this matter to be trivial. Since the pioneering work on peats by Waksman in the 1930s, it is unnecessary to repeat it.
In general, I believe that the conclusions section would be significantly improved if the length were reduced by half, as part of this section includes speculations that do not directly arise from the work itself.
•
Author Response
Dear reviewer!
Thank you for recommendations, provided, below you can see our replies and comments,
Sincerely yours,
Corresponding authors,
Evgeny Abakumov,
Professor, Dept. Applied Ecology, Saint-Petersburg State University, Russia
A report for Molecules-3836039, entitled “Depth-Dependent Molecular Composition of Peat Organic Matter Revealed by 13C NMR Spectroscopy in the Mukhrino Carbon Supersite (Khanty-Mansi region, Russia)”
Abstract:
It is to be regretted that the summary did not provide more detailed information on the type of peat under study, including, for example, the degree of decomposition (sapric, hemic, fibric), or whether it was from a highland or lowland location, or whether it was oligotrophic or mesotrophic, or using classification from the USDA or FAO. In general, biogeochemists attach great importance to knowing whether peat has been formed from lignified vascular plants, or from vegetation lacking lignin, for example mosses.
Reply: the characteristic of turf has been added.
Line 12: “ This study investigates the molecular composition and carbon stabilization mechanisms -- It is unclear whether this paper, which is primarily based on nuclear magnetic resonance (NMR), can provide any insight into "carbon stabilization mechanisms". Instead, the study investigates molecular composition e.g., "for its influence on carbon stabilization mechanisms", or similar.
Reply: corrected
Line 17: “ The study highlights the vulnerability of surface peat layers to mineralization under climate change,” -- In my opinion, this phrase constitutes an unnecessary endorsement of the alleged global change. The surface horizons of all peatlands are always more vulnerable to atmospheric agents because gas exchange with the atmosphere is more active there.
Reply: the phrase has been modified.
Introduction:
Line 35: Some scientific journals require the use of units of the international system (SI), for example, using Mg (megagrams) instead of tons.
Reply: corrected.
Line 51: Please ensure that all references are cited in the text using consecutive numbers in brackets.
Thank you, corrected.
Line 64: “It has been emphasized that aromaticity negatively correlates with aliphaticity for various peat materials in Srednee Priobye (Khanty-Mansiysk region) and that 13C NMR spectroscopy is an informative tool for identifying peat genesis“ -- In my opinion, this sentence is both trivial and tautological. It should either be eliminated or replaced with something more appropriate. It is clear that the different forms of carbon can only be aromatic or aliphatic.
Reply: the phrase was modified.
Line 56: “The author also suggested that increased aliphaticity of humic acids is caused by the woody origin of plant remnants. In contrast, grass peat materials contain relatively higher proportions of aromatic carbon species [10]. -- I have some reservations about the author's interpretation of the findings. I would therefore advise a certain degree of caution. It has long been hypothesised that the lignin of herbaceous plants is more readily biodegradable due to its high cinnamic acid content, which is linked to hemicellulose via ester bonds. Conversely, the lignin of woody plants has been found to exhibit a higher guaiacyl/syringyl ratio, a property that contributes to its enhanced resistance to degradation. It is my belief that the enhanced aromaticity detected in herbaceous vegetation may be attributable to factors other than the nature of the initial material. This is predicated on the premise that the biogeochemical cycle is more active due to the enhanced biodegradability of plant remains, which consequently leads to increased maturity after the accumulation of recalcitrant aromatic compounds. This is a standard occurrence when comparing forest soils with agricultural or deforested soils, where herbaceous remains predominate.
Reply: thank you, we corrected text.
Lines 68 (and 302), (minor): Use M-dash instead of hyphen
Reply: thank you, corrected.
Lines 70, 89, 93, 134… (minor) superscript in 13C
Reply - corrected
Lines 72 and 73 (minor): It may not be necessary to mention the initials in addition to the surname, especially when dealing with such well-known authors.
Reply - corrected
Line 67; “Due to the limited knowledge about molecular methods for stabilizing soil organic matter, particularly in the vast macroareal of Western Siberia,“ -- I would replace "methods" with mechanisms, processes, constraints…
Reply – thank you, corrected
Material and methods
Line 97: Repeated “conifer/ous…”
Reply - corrected
Figure 1: It would be advantageous to include the names of significant cities. This may also be of relevance to nations on the general map. This would be the case for major cities such as St. Petersburg, Moscow, Kazan and Omsk. Similarly, for map B, it is not explained what the blue area around the red circle and the elongated rectangle near Kazan are.
Reply – figure was corrected
Table 1: Please indicate the units for all values given (e.g., %, or preferably (multiplying by 10) g/kg). Reply – I agree that in case of Soil organic carbon it better to use g/kg, but in case of ash and water its more traditional to use %
Please ensure that the N-dash is always used for numerical ranges. Reply - corrected
Please use a decimal point instead of a comma. Reply - done
Peat should be replaced with plant species. Reply – thank you, correcred.
Please note that Table 1 mentions the plant "angustifolium" on two occasions. It is evident that this refers to Sphagnum angustifolium, therefore it should be indicated by the genus and species names (with a lowercase letter) or abbreviated as S. angustifolium in subsequent instances. Similarly, on line 156, the genus name should be abbreviated as S., not Sph.
Thank you, corrected
The same applies to Sphagnum fuscum, which is also referred to as "Fuscum" in the Table.
Thank you, corrected
Please check the use of acronyms carefully: the term "humic acid" is used for the first time on line 48, where the acronym "HAs" should be introduced. However, throughout the manuscript, the complete words are used again on lines 57, 61, 70, 77, etc. Please note that the acronym is introduced on line 126, and the complete words are used again on lines 267, 270, etc.
Thank you, corrected
Please refer to Table 1 for details on whether 'water content' refers to the maximum water-holding capacity at atmospheric pressure or to field humidity. Thank you, clarified
Lines 124 to 129: It is evident that oxygen is calculated using the difference between the three major elements. Thank you, corrected
However, the text does not state whether the percentages of these three elements have been corrected with respect to the ash-free sample. Although the ash content is not too high in these samples, if the percentages are not corrected with respect to the ash-free sample, the error accumulates in oxygen, and the percentages of the other three elements are also incorrect (underestimated). Yes, data was ash corrected, added to the text.
Line 136: Please use the subscript in ZrO2. In the remainder of the page, where numerous functional groups or chemical radicals are indicated, the subscript numbers are also absent. Thank you, corrected.
Line 339: “by the sum of chemical shifts at 0–47 ppm (methyl groups (CH3), methylene groups of long-chain alkyl chains (CH2), and methylene groups of branched alkyl chains (CH, C))” -- It is important to note that a common issue with 13C NMR is the overlap of signals produced by proteins with other significant regions of the spectrum. This overlap primarily occurs in the carbonyl and methoxyl groups, as well as the polymethylene alkyl groups. While the first two cases are referenced in this paper, the influence of peptides (α-amino) in the prominent alkyl group region is not indicated. I would describe this region, for example, as: "0–46 ppm = alkyl + α-amino (13 = methyl, 21 = acetate, 33 = polymethylene)". Thank you, corrected
Could you please indicate how BD (bulk density), AC (ash content %) and WC (water content %) were determined in the material and methods section, which focuses mainly on organic elemental analysis and 13C NMR? Reply – data added.
Results and discussion
Line 157. Italicize Ericaceae (family name) as in Table 1. Corrected
Line 159: peat layers (?) Corrected
Line 162: characteristics Corrected
Figure 2 (caption): Although units such as g/cm³ or g/mL are also used to express density, kg/m³ is the SI base unit for density. Corrected.
Line 169: “The hydrological regime is a key factor determining the degree of decomposition and preservation of organic matter in peat deposits. In the studied profile, the upper layers, despite their low humification degree, exhibit high carbon lability due to their exposure to seasonal water level fluctuations and aeration. In contrast, the deeper horizons, maintained under permanent anaerobic conditions, preserve more stable forms of carbon despite their higher aromaticity” -- I believe the information provided is accurate. However, the wording is very general. Peatlands are distinguished by their layered structure, with the surface material being the most recent and the deeper layers representing the oldest. All of the above is of great importance when it comes to justifying the properties, regardless of the water table. Thank you, the phrase is rephrased.
Why “despite their low humification degree, exhibit high carbon lability“? -- Isn't it a contradiction? (despite). Corrected.
Line 182: “The 182 lowest carbon content, approximately 42%, was found in the deepest layer of the soil section” -- Such low carbon contents have recently been reported for polygonal frozen bogs in Western Siberia [24]“ -- Indeed, the carbon percentage is slightly lower than expected, particularly in comparison to the upper horizons. Corrected.
Despite the absence of any explicit mention to the contrary, it is assumed that the values for the four elements have been corrected based on an ash-free sample. Conversely, it would not be unexpected that the percentage of carbon (C) would be the lowest in the oldest horizon, where the ash content is the highest. Furthermore, the figure indicates that the oxygen content (or O+S+ash?) is higher in the deep horizon, where there is a greater quantity of ash. Reply – data were corrected on ash.
Line 200 and below: “The elevated nitrogen content in deeper layers likely reflects mineral admixture in these eutrophic deposits. These authors also reported a C/N ratio variation of 39–85 (Figure 4). In our study, some horizons exhibit a maximum C/N ratio of up to 150, indicating that the organic matter in our peat column is less enriched in nitrogen compared to the peat material investigated by Raudina and Loiko [25] in permafrost-affected bogs of Western Siberia”-- In this paragraph, two important recommendations are put forward: firstly, the use of tautologies or trivial comments should be avoided; if the carbon-nitrogen ratio is higher, it is evidently because peat has less nitrogen. Reply – text has been modified.
Conversely, and with the requisite caution, hypotheses may be advanced regarding the potential correlation between variations in the C/N ratio and the distinct spatial or temporal impact of hydromorphism. It is important to recall that, in the French school's classification of classic humus types, “peat” and “anmoor” are studied separately from other humus types formed in terrestrial environments: The terms "mull", "moder", and "mor" are employed in this context. This phenomenon can be attributed to the consistently lower C/N ratios observed in hydromorphic soils compared to their terrestrial counterparts. In the former case, this does not indicate greater biological activity, but simply the fact that protein is less easily biodegradable in the hydromorphic environment. Reply – thank you for this idea.
Consequently, and always with caution, an attempt could be made to explain variations in C/N ratio by factors such as the different levels of the water table or the different seasonality of hydromorphism. It is important to note that other factors may also have had an impact, although it should be noted that these are entirely speculative and should not be mentioned without proof. One such factor could be the presence of fires in the past, which have been known to produce an increase in heterocyclic nitrogen forms. These forms are extremely resistant to degradation. Reply – thank you for this idea. Text has been modified.
Lines 216 and below: “One might expect the degree of oxidation to increase with decreasing hydrogenation, but this is not the case. Instead, there is degradation of alkyl groups without the formation of oxidized groups, such as carboxyl groups“ -- On the one hand, the van Krevelen diagram would be discussed with reference to different proportions of alkyl, or aliphatic, and aromatic constituents, rather than hydrogenation or dehydrogenation processes. Conversely, it is not considered necessary to seek singular interpretations regarding the O/C ratios in the van Krevelen diagram. The evolution observed in the graph is logical and expected for humic-type materials with different degrees of maturity.
Indeed, the youngest material in the surface layers has a higher content of aliphatic constituents, and the material with greater degrees of maturity or diagenesis in the deeper layers is the one that presents greater aromaticity. The necessity to seek alternative explanations for the decline in oxygen content with depth is rendered moot by the classical trend towards maturity, discernible in van Krevelen diagrams, characterised by the progressive convergence of the coordinates of increasingly mature samples along the primary diagonal, converging towards the coordinate axis.
This trend is explained as the biomass constituents with the highest oxygen content are carbohydrates. As these carbohydrates degrade preferentially in comparison to lignin or polyphenols, which are aromatic in nature, the oxygen content of the resulting material progressively decreases.
On the other hand, it has been demonstrated that the process of humic acid formation is accompanied by an increase in carboxyl groups. However, in hydromorphic soils, this tendency is not necessarily evident in the van Krevelen diagram applied to whole peat material.
It has been extensively documented in classical literature, e.g., the classical book by Durand: Kerogen: An Insoluble Organic Matter Derived from Sedimentary Rocks. In addition to establishing a correlation between aliphaticity and aromaticity with maturity, he advances a series of other observations on the impact of hydromorphism, distinguishing between terrestrial and aquatic materials, as well as materials of mixed origin, once again based on their position in the van Krevelen diagram. (However, the application of this theory within the same deposit where the effects of hydromorphism and maturation time completely overlap remains unclear).
Finally, the O/C ratio in the sample at the greatest depth (470-490 cm) appears to be significantly elevated, as it also exhibits the highest ash content.
-Thank you! We fully agree with your points regarding the interpretation of the van Krevelen diagram and the molecular evolution of peat organic matter. In response, we have revised the relevant section of the manuscript (Section 3.2) to align with the classical model of organic matter maturation.
In the absence of verification confirming the presence of an error, which is not necessarily an experimental error but rather a calculation error, as previously hypothesised, this outcome would be anticipated. In the event of the carbon, hydrogen, nitrogen and oxygen percentages not being ash-free, it can be deduced that the used carbon, hydrogen and nitrogen values would be lower, whilst the percentage of oxygen would be exaggeratedly higher. Reply – elementary composition was ash corrected which we calculated Oxygen and elementary ratios.
Figure 4: caption: Krevelen
Thank you! The caption has been corrected.
Apparently, there are two Fig. 5.
Figure 5 (former): The term "turf" is used in the figure caption instead of "peat."
Thank you! It was corrected.
In addition, the decline of the C/N ratio is a very relevant indicator, as it is known to decrease progressively in the deepest layers, which are presumed to be most affected by hydromorphism. I do not consider it necessary to present graphs for the H/C and O/C atomic ratios in bar form. Furthermore, I always recommend checking that the values are calculated using an ash-free sample. If this information were available, many of the graphs presented could show the upper, lower, or average limits of the water table, if it varies substantially throughout the year (which, incidentally, is not very detailed in the text).
Thank you! The additional information has been added to 3.2 Section.
Line 232: “ This indicates that the depth 232 of the sample and the degree of organic matter transformation are more important than 233 the peat composition. ” -- It is my understanding that in this sentence, the authors are not referring so much to the composition of the peat as to the chemical composition of the original vegetation. – thank you, corrected.
Line 245: “In this case, humification appears to be driven by proton loss rather than intensive oxidation of organic matter“ -- In line with my previous recommendations, I suggest using terms such as "aliphatic content" or "aromatic content," or "aromatization" or "selective degradation of aliphatic or O-alkyl constituents," where appropriate. – thank you, corrected.
In addition, it is my professional opinion that the observed trends are indicative of classical peat maturation. The absence of a progressive increase in oxygen content should not be regarded as an abnormality. It is important to note that the process being studied involves working with complete peat and not with humic acids. The decrease in oxygen is expected from the selective degradation of carbohydrates, which represent the majority of plant biomass. This is particularly evident in peat dominated by Sphagnum, a lignin-lacking plant. – thank you for this comment.
Figure 5 (latter): Improve the proportions of the lettering, especially in the x-axis labels.
Corrected (with corrections of figures numbers
Caption: 13C (with superscript) instead of 13-C
Thank you! It was corrected.
With regard to the spectra, I generally prefer to normalise them by height or area, although this is a relatively minor detail. Thank you, if possible, we leave at the current form.
Table 2
Please use decimal dots instead of commas.
Please use N-dash for numerical ranges (see first column and first row for ppm ranges).
Thank you, done.
I consider it useful that, although it has already been briefly explained, some words be indicated to illustrate the meanings of the different chemical shift ranges, for example alkyl, O-alkyl, H-aromatic, C-N-aromatic...
Line 251 and below ” In general, a low degree of humification is characteristic of peats due to low aeration and high moisture content [30].” -- While the aforementioned points are accurate, it is advisable to exercise caution when writing, as there are two overlapping factors that influence maturity depending on depth (hydromorphy and time). The deep layers are the most affected by hydromorphy; however, they are also the oldest, and therefore exhibit the highest degree of diagenesis. – Thank you, text has been modified.
Line 291: “This suggests that if peat layers currently not exposed on the surface were to become exposed, active humification would likely occur“ -- Whilst there may be some veracity in this claim, it is a rather general comment that would benefit from further clarification. It may be preferable to avoid mentioning this issue, given that peats are hydromorphic soils where the degree of humification can reach advanced levels after several centuries without being exposed to active gas exchange with the atmosphere. The formation of ITS humic substances is a gradual process that requires a significant amount of time.
Following peatland desiccation and the lowering of the water table, the degree of humification typically increases. However, mineralisation predominates, with peats spontaneously combusting without flame. This process has destructive yet intense effects on the composition of the peaty material. Thank you, the sentence has been corrected.
Conclusions
Line 300: “Given the demonstrated lability of surface peat layers and their disproportionate sensitivity to temperature fluctuations our molecular characterization suggests these systems may exhibit nonlinear responses to warming” -- Please be advised that the following applies: I would advise against making speculative statements regarding the alleged effect of temperature. The effect of temperature is highly dependent on humidity, particularly in relation to whether the temperature increases during the cold or warm seasons of the year. However, the primary determining factor is the duration of the desiccation periods of the surface layers. Furthermore, the potential changes in vegetation types over time, in combination with the historical impact of fire or the contribution of surface materials, significantly complicate the reconstruction of the processes of organic matter formation in peatlands. This suggests that conclusions should be drawn with caution. Thank you – sentence modified.
Line 311: “(1) humification indices can be reliably obtained without alkaline extraction“ -- I consider this matter to be trivial. Since the pioneering work on peats by Waksman in the 1930s, it is unnecessary to repeat it. Corrected.
In general, I believe that the conclusions section would be significantly improved if the length were reduced by half, as part of this section includes speculations that do not directly arise from the work itself.
Reply: Conclusion was reduced
Round 2
Reviewer 1 Report
Comments and Suggestions for Authors
Dear Authors,
Thank you for improving your manuscript "Depth-Dependent Molecular Composition of Peat Organic Matter Revealed by ¹³C NMR Spectroscopy in the Mukhrino Carbon Supersite (Khanty-Mansi region, Russia)". Reading carefully the new version of this work, I observed that the conclusions section contain several overstatements based on the data presented in the manuscript. Therefore I recommend for authors to be very careful with statements as "bulk-sample NMR approach resolved methodological debates". For me this is a huge overstatement because the study did not presents well comparative analysis with established methods, validation against reference standards, and systematic evaluation of methodological uncertainties that would support such a bold claim.
Overall the conclusion section present the findings with inappropriate certainty given the methodological limitations, missing error propagation, and absence of validation studies, therefore I recommend for authors to carefully reconsider this section.
Comments on the Quality of English LanguagePlease very the grammar
Author Response
Deae reviewer!
Thank You for you reccomendation to make coclusions chapter more precise, we have modified it according to Your suggestions,
Sincerely Yours,
Evgeny Abakumov, corresponding author,
professor, Saint-Petersburg State University